# Quantum-Amplitude Embedded Adaptation for Parameter-Efficient Fine-Tuning in LLMs

Emily Jimin Roh[0009−0008−0013−6342] and Joongheon Kim[0000−0003−2126−768X]

Korea University, Seoul 02841, Korea
emilyjroh@korea.ac.kr joongheon@korea.ac.kr

**Abstract.** Large language models (LLMs) require substantial resources for task-specific adaptation, that motivates the development of parameter-efficient fine-tuning (PEFT) methods. This paper presents quantum-amplitude embedded adaptation (QAA), a novel PEFT framework that logarithmically compresses activation vectors using quantum-amplitude embedding and applies expressive non-linear transformations via parameterized quantum circuits (PQCs). By replacing linear adapters in attention modules with compact quantum modules, QAA achieves high expressivity while drastically reducing the number of trainable parameters. Empirical results demonstrate that QAA performs on par with or better than existing PEFT under constrained memory and compute budgets, highlighting its potential for efficient LLM fine-tuning.

**Keywords:** Quantum Computing· Large Language Model

## 1 Introduction

**Background and Motivation.** Large language models (LLMs) have demonstrated remarkable performance across diverse natural language processing (NLP) tasks, including summarization, question answering, and instruction following (Neumann et al., 2025; Zhou et al., 2024a). However, to fully leverage this capability for domain-specific generation or structured prediction tasks, additional fine-tuning on downstream data remains essential (Gao et al., 2025; Zhang et al., 2024a). Fine-tuning allows the model to specialize its behavior by conditioning on task-relevant data, thereby improving performance on target distributions that differ from the pre-training corpus (Ahn et al., 2025; Wu et al., 2025b; Xu et al., 2024). Despite its effectiveness, full fine-tuning of LLMs remains computationally prohibitive due to their enormous parameter sizes (Kasneci et al., 2023; Lin et al., 2025). Updating all parameters for each downstream task becomes impractical, particularly in multi-task or resource-constrained settings (Schmirler et al., 2024; Yang et al., 2024). To address this, parameter-efficient fine-tuning (PEFT) strategies have been proposed (Ali et al., 2025; Ma et al., 2022). Among the various strategies, low-rank adaptation (LoRA) (Hu et al., 2022) is introduced to reduce the cost of fine-tuning LLM by injecting trainable low-rank matrices into frozen attention and feed-forward layers, effectively constraining

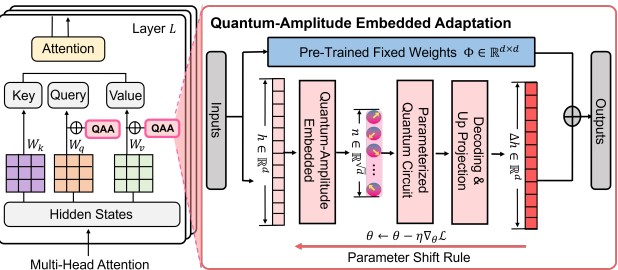

Fig. 1: The overall architecture of QAA.

updates to a smaller parameter subspace. Building on the principle of minimal parameter modification, prefix tuning (Huang et al., 2023) proposes an alternative approach that prepends a small number of trainable vectors to the input sequence, conditioning the model's behavior without altering its internal weights. However, existing PEFT techniques predominantly utilize linear projection layers or discretized prompt tokens, which inherently constrain their representational capacity (Kim et al., 2025; Wu et al., 2025a). Such approaches are insufficient to capture the non-linear, compositional, and context-sensitive transformations required for effective task adaptation (Lv et al., 2024; Zhou et al., 2024c). This limitation becomes more pronounced under extremely tight parameter budgets, where a trade-off emerges between fine-tuning efficiency and representational expressivity (Chang et al., 2024).

**Algorithm Concept.** To address the limitations of existing PEFT methods, this paper introduces the quantum-amplitude embedded adaptation (QAA), a novel quantum parameterized modules into transformer-based architectures by replacing classical linear adapters with a structure that combines logarithmic amplitude embedding (Cuéllar et al., 2023; Gonzalez-Conde et al., 2024) and parameterized quantum circuit (PQC) (Baek et al., 2023; Mahmud et al., 2025) as illustrated in Fig. 1. This architecture enables a compressed representation of the input vector in logarithmic space while enhancing transformation expressivity through quantum operations (Kottahachchi Kankanamge Don and Khalil, 2025). The compressed representation is then processed by a PQC, which performs non-linear transformations over entangled qubit states. These transformations enable the modeling of complex input-output mappings that are difficult to express with conventional low-rank or quantized layers. Therefore, QAA aims to preserve performance while minimizing the adaptation overhead. This design bridges the gap between efficiency and expressivity in PEFT, offering a new direction for scalable fine-tuning under resource constraints.

**Contributions.** First of all, the novel PEFT framework, QAA is proposed, which enables logarithmic compression via quantum-amplitude embedding and expressive non-linear adaptation via PQC. In addition, the proposed method integrates seamlessly with transformer-based LLMs without modifying pretrained weights, ensuring modular and scalable deployment. Lastly, experimental results show that QAA achieves competitive performance compared to classical baselines under constrained parameter settings.

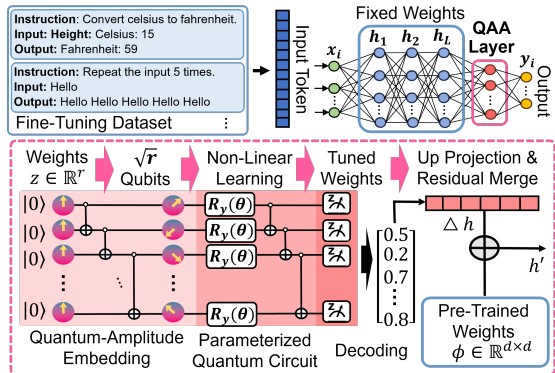

Fig. 2: QAA-based LLM fine-tuning framework.

## 2  LLM Fine-Tuning Framework

The LLM fine-tuning is computationally expensive due to their massive parameter sizes, often ranging from hundreds of millions to billions (Ding et al., 2023; Kalyan, 2024). Updating full model copies for each downstream task is impractical, especially in multi-task, low-resource, or on-device scenarios with limited memory and compute (Lian et al., 2022; Zhou et al., 2024b). Let $\mathcal{Z} = \{(x_i, y_i)\}_{i=1}^{N}$ denote the task-specific dataset, where $x_i$ is an input sequence (e.g., a prompt, document, or question) and $y_i$ is the corresponding output sequence. Given a pre-trained autoregressive language model $P_\Phi(y \mid x)$, where $\Phi \in \mathbb{R}^{|\Phi|}$ denotes the full set of model parameters and $|\Phi|$ is the total number of tunable weights, the standard fine-tuning objective maximizes the conditional log-likelihood of the target sequence $y$ given the input $x$, as follows,

$$\max_{\Phi} \sum_{(x,y)\in\mathcal{Z}} \sum_{t=1}^{|y|} \log P_\Phi(y_t \mid x, y_{<t}). \tag{1}$$

In full fine-tuning, all parameters $\Phi$ are updated, resulting in a unique fine-tuned model for each task (Liu et al., 2023). However, as $|\Phi|$ becomes large (e.g., over 175 billion for GPT-3), this becomes infeasible in memory footprint, training time, and deployment cost (Brown et al., 2020; Hoffmann et al., 2022).

To address this, PEFT methods have been developed to adapt only a small subset of parameters while keeping the majority of $\Phi$ frozen (Usman et al., 2024). Classical PEFT techniques such as LoRA (Mao et al., 2025), insert lightweight trainable modules into the LLM attention or feedforward blocks. Formally, PEFT methods aim to learn a low-dimensional update function $\Delta h(\theta)$, parameterized by $\theta \in \mathbb{R}^{|\theta|}$ where $|\theta| \ll |\Phi|$, such that the model parameters become $\Phi + \Delta h(\theta)$ (Thomas et al., 2024). The fine-tuning objective is then rewritten as,

$$\max_{\theta} \sum_{(x,y)\in\mathcal{Z}} \sum_{t=1}^{|y|} \log P_{\Phi+\Delta h(\theta)}(y_t \mid x, y_{<t}). \tag{2}$$

We define $\Delta\Phi(\theta)$ as a quantum-amplitude adapter that logarithmically compresses the full activation $x \in \mathbb{R}^d$ via amplitude embedding and applies trainable quantum circuits for efficient fine-tuning.

Table 1: Comparison of parameter complexity across PEFT.

| Method | Trainable Parameters |
|---|---|
| Full Fine-Tuning | $\mathcal{O}(d^2)$ |
| LoRA (rank-$r$) | $\mathcal{O}(dr)$ |
| Prefix Tuning (length-$l$) | $\mathcal{O}(ld)$ |
| **QAA (proposed)** | $\mathcal{O}(d \log d)$ |

## 3   Algorithm Design

### 3.1   QAA

This paper proposes the QAA, a novel PEFT mechanism that integrates quantum-amplitude embedding and PQC into the LLM adaptation process. As shown in Fig. 2, QAA replaces conventional full-rank linear adapters by projecting hidden activation vectors into a quantum feature space. This approach enables expressive, non-linear transformation while reducing the number of trainable parameters to a logarithmic scale with respect to the hidden dimension $d$.

**Quantum-Amplitude Embedding** Let $x_i \in \mathbb{R}^d$ denote the hidden activation vector of a pretrained language model, such as the output of an attention sub-layer. To represent this vector in a quantum system, we employ amplitude embedding, which encodes a real-valued vector into the amplitudes of a quantum state. The number of qubits $n$ is selected such that $n = \lceil \log_2 d \rceil$, ensuring that the quantum state can represent a vector of dimension $2^n$. To align the dimension with quantum requirements, the number of qubits $n$ is selected such that $n = \lceil \log_2 d \rceil$. Then, $x_i$ is transformed into $x_i' \in \mathbb{R}^{2^n}$ by zero-padding or truncating to $x_i' \in \mathbb{R}^{2^n}$. The input vector $x_i'$ is normalized as, $\tilde{x}_i = \frac{x_i'}{\|x_i'\|_2}$. Where the $\ell_2$ norm is defined as $\|x_i'\|_2 = \sqrt{\sum_{j=0}^{2^n-1}(x_{i,j}')^2}$. This step guarantees that the amplitude embedded quantum state satisfies the normalization constraint $\langle x_i | x_i \rangle = 1$ ensuring that the total probability amplitude across all basis states sums to one. Then, the normalized vector $\tilde{x}_i$ is embedded into a quantum state, $|x_i\rangle = \sum_{j=0}^{2^n-1} \tilde{x}_{i,j} |j\rangle$, where each component $\tilde{x}_{i,j}$ becomes the amplitude of basis state $|j\rangle$ in the quantum system. This representation enables the quantum circuit to operate on the entire input vector in superposition, facilitating global, parallel processing. Amplitude embedding enables logarithmic compression of input vectors, requiring only $\log_2 d$ qubits to represent a $d$-dimensional activation vector. This compact encoding not only reduces the number of required qubits but also allows the PQC to operate on a superposition of features, enabling efficient learning of complex, entangled representations.

**PQC-based Non-Linear Learning** After amplitude embedding, the quantum state is processed by a PQC to enable expressive, non-linear transformations in

a high-dimensional Hilbert space (Khairy et al., 2020). Let the input quantum state be $|x_i\rangle \in \mathbb{C}^{2^n}$ across $n$ qubits. The PQC applies a sequence of parameterized gates followed by entangling operations, forming a unitary transformation $U_T(\theta)$, where $\theta \in \mathbb{R}^n$ denotes the trainable parameters.

The PQC consists of two main operations. First, for each qubit $j \in \{1, \ldots, n\}$, a parameterized single-qubit rotation gate $R_Y$ is applied, defined as, $R_Y(\theta_j) = \exp\left(-i\frac{\theta_j}{2}Y\right)$, where $Y$ is the Pauli-Y matrix and $\theta_j$ denotes the learnable parameter for qubit $j$. These rotations introduce controlled non-linearity at the qubit level, enabling the circuit to represent more complex transformations than classical linear layers. Second, to model inter-qubit dependencies, the circuit includes a fixed entanglement pattern using CNOT (controlled-NOT) gates. For adjacent qubits $(j, j+1)$, the CNOT operation acts as,

$$\text{CNOT}_{j,j+1} |a\rangle_j |b\rangle_{j+1} = |a\rangle_j |a \oplus b\rangle_{j+1}, \tag{3}$$

where $\oplus$ denotes bitwise XOR. These entangling operations create quantum correlations across the qubits, allowing the PQC to model joint dependencies and capture task-specific interactions (Pappalardo et al., 2025).

**Decoding and Up Projection** After the PQC processes the amplitude embedded quantum state, the resulting quantum state $|\psi_i\rangle \in \mathbb{C}^{2^n}$ is measured in the computational basis. To extract classical features from this quantum state, we perform expectation-value measurements with respect to Pauli-Z observables on each qubit. Specifically, the output vector is defined as,

$$z_i^{(q)} = [\langle \psi_i | Z_j | \psi_i \rangle]_{j=1}^n \in \mathbb{R}^n, \tag{4}$$

where $Z_j$ denotes the Pauli-Z operator acting on the $j$-th qubit, and $\langle \psi_i | Z_j | \psi_i \rangle$ is the expected measurement outcome on that qubit. This vector $z_i^{(q)}$ serves as a compact representation of the task-relevant quantum features encoded during the PQC transformation.

To reintegrate the quantum features into the original LLM architecture, we apply a learnable linear transformation, or up-projection, via a matrix $W_u \in \mathbb{R}^{d \times n}$, where $d$ is the hidden dimension of the base model. The resulting decoded activation is given by, $\hat{x}_i = W_u z_i^{(q)} \in \mathbb{R}^d$. Finally, this decoded activation is combined with the frozen hidden state $x_i$ through a residual merge, $x_i^{(\text{aug})} = x_i + \hat{x}_i$.

This complete process—including amplitude embedding, PQC transformation, quantum measurement, and up-projection—defines the update function $\Delta\Phi(\theta)$, which enables PEFT with logarithmically fewer learnable parameters compared to full model adaptation.

### 3.2   Complexity and Expressivity Analysis

Classical PEFT methods, such as LoRA and prefix tuning, reduce the number of trainable parameters by restricting the update space. In LoRA, a full-rank

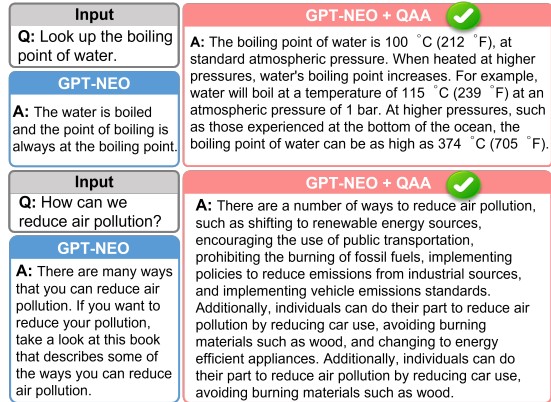

Fig. 3: Example output generated using QAA-based LLM.

Table 2: Comparison of the standard NLG evaluation metrics using different adaptation methods.

| Model & Method | #Model Parameters | #Trainable Parameters | Trainable Ratio | Standard NLG Evaluation Metrics | | | | |
|---|---|---|---|---|---|---|---|---|
| | | | | BLEU | ROUGE-1 | ROUGE-2 | ROUGE-L | BERTScoreF1 |
| GPT-NEO (Full) | 125.2M | 125.2M | 100% | 10.62 | 18.65 | 17.28 | 18.35 | 83.06 |
| GPT-NEO (LoRA) | 125.2M | 294.9K | 0.24% | 0.734 | 7.179 | 2.069 | 5.874 | 77.76 |
| GPT-NEO (Prefix) | 125.2M | 368.6K | 0.29% | 0.827 | 8.585 | 2.169 | 6.833 | 76.95 |
| **GPT-NEO (QAA)** | 125.2M | **184.5K** | **0.14%** | **33.37** | **47.58** | **45.27** | **46.51** | **89.11** |
| TinyLLaMA (Full) | 1.100B | 1.100B | 100% | 24.25 | 25.06 | 24.34 | 25.06 | 85.44 |
| TinyLLaMA (LoRA) | 1.100B | 1.126M | 0.11% | 12.24 | 21.14 | 20.64 | 21.14 | 85.13 |
| TinyLLaMA (Prefix) | 1.100B | **0.225M** | **0.02%** | 21.07 | 12.37 | 21.07 | 24.88 | 87.17 |
| **TinyLLaMA (QAA)** | 1.100B | 0.540M | 0.05% | **20.66** | **40.28** | **32.13** | **36.29** | **88.17** |
| Qwen2.5 (Full) | 494.0M | 494.0M | 100% | 13.92 | **21.90** | 21.30 | **21.88** | 86.14 |
| Qwen2.5 (LoRA) | 494.0M | 0.540M | 0.10% | 6.549 | 15.45 | 12.94 | 17.99 | 83.78 |
| Qwen2.5 (Prefix) | 494.0M | **0.122M** | **0.02%** | 1.715 | 14.08 | 4.984 | 9.229 | 82.56 |
| **Qwen2.5 (QAA)** | 494.0M | 0.215M | 0.04% | **12.95** | 21.77 | **21.56** | 21.76 | **87.76** |

matrix update is decomposed as $W \leftarrow W + AB$ with $A \in \mathbb{R}^{d \times r}, B \in \mathbb{R}^{r \times d}$, yielding a parameter cost of $\mathcal{O}(dr)$. Prefix tuning prepends $l$ trainable vectors of dimension $d$, resulting in $\mathcal{O}(ld)$ parameters. While effective, these methods are inherently linear and limited in expressivity.

The proposed QAA departs from this linear paradigm. It performs amplitude embedding of the hidden activation $x_i \in \mathbb{R}^d$ into a quantum state $|x_i\rangle \in \mathbb{C}^{2^n}$, with $n = \lceil \log_2 d \rceil$. The transformation is carried out by a trainable PQC with $L$ layers acting on $n$ qubits. This circuit introduces $\mathcal{O}(L \cdot \log d)$ parameters. The expectation values from the PQC output are decoded by an up-projection matrix $W_u \in \mathbb{R}^{d \times \log d}$ to match the original hidden size. Hence, the total trainable parameter count is bounded by, $|\Delta \Phi|_{\text{QAA}} = \mathcal{O}(d \log d)$. Compared to LoRA's $\mathcal{O}(dr)$ and prefix tuning's $\mathcal{O}(ld)$, QAA achieves lower parameter complexity when $r, l = \Omega(\log d)$. Moreover, PQCs enable expressive non-linear transformations and qubit entanglement, which allows QAA to approximate complex functions beyond the capacity of linear modules.

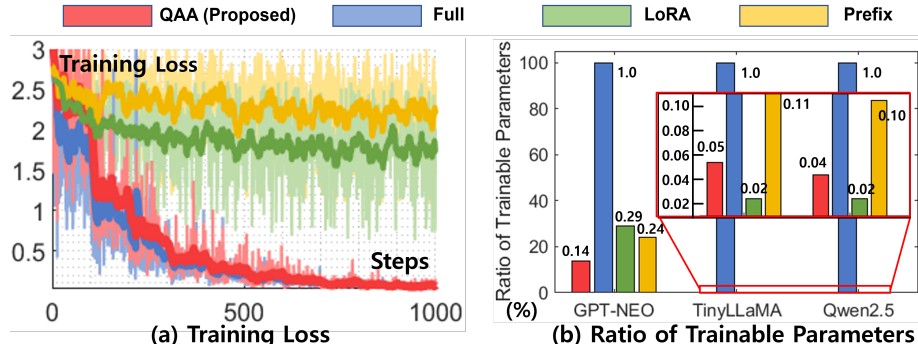

Fig. 4: Training loss comparison across 1,000 steps and ratio of trainable parameters (%) across models.

## 4   Performance Evaluation

To validate the feasibility of the proposed QAA, experiments are conducted across three LLM models—GPT-NEO (Black et al., 2021), TinyLLaMA (Zhang et al., 2024b), and Qwen2.5 (Xiang et al., 2025)—on the standard NLG evaluation metrics on Alpaca (Taori et al., 2023) dataset. The evaluation compares QAA with baselines: full fine-tuning, LoRA with $r = 8$, and prefix tuning. Performance is assessed in terms of BLEU, ROUGE (1, 2, L), and BERTScore F1, alongside the number and ratio of trainable parameters.

**Qualitative Evaluation.** Fig. 3 presents example responses comparing GPT-NEO and GPT-NEO + QAA on instruction-following tasks. In both cases, the QAA-enhanced model produces answers that are significantly more informative and context-aware. In the boiling point query, GPT-NEO provides a vague and partially correct answer, while GPT-NEO + QAA offers a detailed explanation including scientific units, conditions (e.g., atmospheric pressure), and contextual scenarios. Similarly, in the air pollution task, the base model defers the answer to an external resource, whereas the QAA-augmented model provides actionable strategies, demonstrating improved reasoning and informativeness. These validate QAA's ability to enhance the LLMs with minimal additional parameters.

**Training Efficiency.** Fig. 4(a) illustrates the training loss curve over 1000 optimization steps across four fine-tuning strategies. The proposed QAA exhibits the fastest convergence rate, achieving a training loss below 0.5 within approximately 300 steps. In contrast, prefix tuning and LoRA show slower and less stable convergence behaviors. While full fine-tuning shows a convergence rate comparable to that of QAA. These results indicate that QAA not only accelerates convergence but also achieves the lowest final training loss among all PEFT methods. The improvement is attributed to the expressive power of the quantum circuit, enhanced function approximation, despite utilizing fewer trainable parameters.

**Parameter Efficiency.** Fig. 4(b) and Table 2 compare the ratio of trainable parameters across different fine-tuning methods and model backbones. The pro-

posed QAA consistently maintains one of the lowest parameter footprints among all baselines. Specifically, for GPT-NEO, QAA updates only 184.5K parameters, corresponding to 0.14% of the full model size, which is significantly lower than LoRA (0.24%) and prefix tuning (0.29%). For TinyLLaMA, QAA trains merely 0.05% of the parameters, compared to 0.11% for LoRA and 0.02% for prefix tuning. Similarly, in the case of Qwen2.5, QAA achieves a trainable ratio of just 0.04%, outperforming LoRA (0.10%) and prefix tuning (0.02%) in parameter efficiency. These results demonstrate that QAA substantially reduces the number of trainable parameters while still maintaining competitive performance.

**Generation Quality.** In Table 2, QAA consistently outperforms LoRA and prefix tuning across all model backbones while maintaining a significantly lower number of trainable parameters. On GPT-NEO, QAA achieves a BLEU score of 33.37 and a BERTScore of 89.11, both of which are substantially higher than those of LoRA (BLEU: 0.734, BERTScore: 77.76) and prefix tuning (BLEU: 0.827, BERTScore: 76.95), and even surpass the full fine-tuning baseline (BLEU: 10.62, BERTScore: 83.06). For TinyLLaMA, QAA achieves a BERTScore of 88.17 with only 0.05% of trainable parameters, approaching the full model's score of 85.44 while also improving upon LoRA and prefix tuning in ROUGE-1 and ROUGE-L. In the case of Qwen2.5, QAA achieves a BLEU score of 12.95 and BERTScore of 87.76, again outperforming LoRA (BLEU: 6.549, BERTScore: 83.78) and prefix tuning (BLEU: 1.715, BERTScore: 82.56), and closely matching the full model's performance. These results indicate that QAA effectively captures both lexical precision and semantic fidelity, achieving state-of-the-art generation quality.

## 5   Conclusion

This paper presents QAA, a novel quantum adapter for PEFT of LLMs. QAA leverages quantum circuit-based non-linear transformations to enhance the expressiveness of low-rank adaptations while significantly reducing the number of trainable parameters. Experiments across LLM backbones, demonstrate that QAA achieves competitive performance to full fine-tuning and consistently outperforms existing PEFT methods. These results highlight QAA's potential as a practical and scalable fine-tuning strategy for resource-constrained environments and downstream applications.

## 6   Acknowledgments

The complete version of this paper is currently under review at the ACM Conference on Information and Knowledge Management (CIKM), Seoul, Korea, October 2025. The corresponding author of this paper is Joongheon Kim (e-mail: joongheon@korea.ac.kr).

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
