# OpenReview forum: "Quantum-Amplitude Embedded Adaptation for Parameter-Efficient Fine-Tuning in LLMs"
_purdue.edu/Purdue_University/PQAI/2025/Symposium — PQAI 2025 Oral_

### Official Review · Reviewer_31Kf · 2025-07-25
**An intriguing quantum-inspired PEFT approach with promising results, yet lacking practical quantum validation**

**Rating:** 6
**Confidence:** 4

**Review:**

The paper introduces QAA, a parameter-efficient fine-tuning framework that embeds hidden activations into quantum-amplitude states and applies parameterized quantum circuits (PQCs) to replace classical adapters in LLMs, achieving logarithmic parameter complexity.

Its main strength lies in marrying quantum compression (amplitude embedding) with expressive non-linear adaptation via PQCs, which yields a compelling trade-off.

Weaknesses: 1) The paper offers no discussion of qubit noise, gradient estimation on hardware, or wall-time overhead of simulating PQCs of depth L. 2) Although parameter counts are low, the memory and latency cost of back-propagating through simulated quantum circuits is not measured, leaving open whether QAA is truly “efficient”. 3) Only a single dataset (Alpaca) and limited model families are tested, and no ablation studies are provided to isolate the contribution of amplitude embedding versus PQC-based non-linearities.

---

### Decision · Program_Chairs · 2025-07-29

Accept (Oral)